# Tellurium-Doped, Mesoporous Carbon Nanomaterials as Transparent Metal-Free Counter Electrodes for High-Performance Bifacial Dye-Sensitized Solar Cells

**DOI:** 10.3390/nano10010029

**Published:** 2019-12-20

**Authors:** Chang Ki Kim, Jung-Min Ji, Haoran Zhou, Chunyuan Lu, Hwan Kyu Kim

**Affiliations:** Global GET-Future Lab. & Department of Advanced Materials Chemistry, Korea University, 2511 Sejong-ro, Sejong 339-700, Korea; rornfl1982@naver.com (C.K.K.); manbbong@korea.ac.kr (J.-M.J.); zhouhaoran@naver.com (H.Z.); lcy99168@gmail.com (C.L.)

**Keywords:** dye-sensitized solar cells, counter electrodes, bifacial devices, tellurium-doped mesoporous carbon, transparency

## Abstract

Tellurium-doped, mesoporous carbon nanomaterials with a relatively high doping level were prepared by a simple stabilization and carbonization method in the presence of a tellurium metalloid. A transparent counter electrode (CE) was prepared using tellurium-doped, mesoporous carbon (TeMC) materials, and was directly applied to bifacial, dye-sensitized solar cells (DSSCs). To improve the performance of the bifacial DSSC device, CEs should have outstanding electrocatalytic activity, electrical conductivity, and electrochemical stability, as well as high transparency. In this study, to make transparent electrodes with outstanding electrocatalytic activity and electrical conductivity, various TeMC materials with different carbonization temperatures were prepared by simple pyrolysis of the polyacrylonitrile-block-poly (n-butyl acrylate) (PAN-b-PBA) block copolymer in the presence of the tellurium metalloid. The electrocatalytic activity of the prepared TeMC materials were evaluated through a dummy cell test, and the material with the best catalytic ability was selected and optimized for application in bifacial DSSC devices by controlling the film thickness of the CE. As a result, the bifacial DSSC devices with the TeMC CE exhibited high power conversion efficiencies (PCE), i.e., 9.43% and 8.06% under front and rear side irradiation, respectively, which are the highest values reported for bifacial DSSCs to date. Based on these results, newly-developed transparent, carbon-based electrodes may lead to more stable and effective bifacial DSSC development without sacrificing the photovoltaic performance of the DSSC device.

## 1. Introduction

Dye-sensitized solar cells (DSSCs), which can directly convert solar energy into electrical energy, are seen as promising energy conversion devices due to their high power conversion efficiency (PCE), easy fabrication process, and environmentally-friendly nature [1,2,3,4]. In addition to these advantages, DSSCs have favorable characteristics such as their color and transparency, as well as their efficiency under ambient light conditions (i.e., for indoor operation) [5]. Therefore, these devices could be applied to building-integrated photovoltaics (BIPVs), which can utilize external and internal lights [6,7].

Conventional DSSC devices consist of the three following main components: a dye-absorbed TiO_2_ photoanode, a redox couple (iodine: I^−^/I_3_^−^, cobalt: Co(bpy)_3_^2+/3+^) [8,9,10], and a counter electrode (CE). Among these main components, the role of the CE in DSSC devices is for electron transfer from the external circuit to the electrode surface for the reduction of the oxidized redox species [11,12,13]. Therefore, CEs should have high electrical conductivity and electrocatalytic activity for high-performance DSSC devices. Generally, platinum (Pt) has been widely used as a CE in DSSCs because of its outstanding electrical conductivity and electrocatalytic activity. Nevertheless, Pt-based CEs cannot be applied to large-scale applications because of their high cost and limited supply. Therefore, it is necessary to develop novel materials at low cost, and with high electrocatalytic activity and electrical conductivity in order to replace Pt. In recent years, various candidate materials such as carbonaceous materials [14,15,16,17], polymers [18,19], metal compounds [20,21] and their combinations [22,23] have been developed to this end. Among them, carbon materials are used in many fields, such as DSSCs [24,25], fuel cells [26], supercapacitors [27], batteries [28], and sensors [29], because of their high electrical conductivity, high specific surface area, and low cost.

In this study, we prepared carbon-based CEs using tellurium-doped, mesoporous carbon (TeMC) materials to replace Pt-based CEs, and applied them to the CEs in DSSCs. Previous studies have already reported that Te-doped carbon materials can be prepared and applied as the CEs of a DSSC, showing outstanding electrocatalytic performance for cobalt and iodine electrolytes [30]. However, due to the low doping level of tellurium in the carbon structure, it did not perform significantly better than Pt. Therefore, we improved the electrocatalytic activity by increasing the doping level of tellurium; at the same time, a transparent electrode was prepared while maintaining its high electrocatalytic activity, which was introduced into the bifacial DSSC device. To make bifacial DSSC devices, the CE and electrolyte must be transparent, because irradiated light from the rear side passes through the CE and the electrolyte before finally reaching the photoanode. Therefore, the CE should have the following characteristics for use in bifacial DSSCs: (i) high electrocatalytic activity for the reduction of oxidized redox couples compared with Pt-based CEs, (ii) high electrical conductivity for the transfer of electrons from the external circuit to the electrode surface, (iii) high transmittance to improve the performance of rear-side irradiation, and (iv) high electrochemical stability for industrial applications [31,32,33]. In this study, to make the transparent electrode while maintaining its electrocatalytic activity and electrical conductivity, we prepared TeMC materials with different carbonization temperatures. According to the carbonization temperature, as-prepared TeMC materials have different Te doping levels and electrical conductivities; therefore, they have different performance when used as CEs. The electrocatalytic abilities of the prepared TeMC materials were evaluated through a dummy cell test, and the material with the best catalytic ability was selected for application in bifacial DSSC devices. To prepare the transparent CE, the selected a TeMC material that was optimized by controlling the film thickness when preparing the CE. The optimized TeMC CE had excellent electrocatalytic activity, electrical conductivity, and electrochemical stability, as well as high transmittance, and it showed excellent performance when applied to the bifacial DSSC device.

## 2. Experimental Section

### 2.1. Materials

N-butyl acrylate, α, α’-azoisobutyronitrile, CuBr_2_, tris(2-pyridylmethyl)amine, ethyl α-bromoisobutyrate, acrylonitrile, 2,2’-bipyridyl, CuCl, Al_2_O_3_, dimethylformamide, dimethyl sulfoxide, ethanol, toluene, methanol, and tellurium were all obtained from Sigma-Aldrich (Seoul, Korea).

### 2.2. Preparation of Tellurium-Doped, Mesoporous Carbon (TeMC) Materials

The PAN-b-PBA (polyacrylonitrile-block-poly (n-butyl acrylate) copolymer (2 g) was physically mixed with 1 g of tellurium powder and placed into a quartz furnace. The mixture was stabilized at 280 °C for 2 h under an air atmosphere (heating rate: 5 °C/min) and then carbonized at three different temperatures (700, 800, and 900 °C) for 1 h under a nitrogen atmosphere (heating rate: 5 °C/min). The final products were washed with 1 M HCl and 3 M KOH aqueous solution to remove the unreacted tellurium.

## 3. Results and Discussion

In a previous study, we successfully prepared the polyacrylonitrile-block-poly(n-butyl acrylate) (PAN-b-PBA) block copolymer using the atom transfer radical polymerization method, and directly used it to make tellurium-doped porous carbon materials [24,30]. The prepared carbon materials were used as a CE in DSSCs, and showed outstanding performance due to their mesoporous structure and many defect sites. In this study, to increase the electrocatalytic activity, we improved the doping level of tellurium by a simple stabilization and carbonization method in the presence of a tellurium metalloid. The preparation method is as follows (see Figure 1). First, the prepared block copolymer (PAN-b-PBA), according to a previous report [30], and tellurium metalloid were physically mixed; then, the mixture was put into a quartz furnace. To prepare the porous carbon materials, the mixture was stabilized under an air atmosphere and carbonized under a N_2_ atmosphere. After these processes, to remove the unreacted elemental tellurium, the resultant materials were washed with 1 M HCl and 3 M KOH aqueous solution. To investigate the effect of temperature during the preparation process, we prepared the TeMC materials at different temperatures (700, 800, and 900 °C) in the carbonization process. The preparation method of TeMC materials is described in the Experimental section.

Field emission scanning electron microscopy (FE-SEM) (Hitachi, Japan) and high-resolution transmission electron microscopy (HR-TEM) (JEOL Ltd., Japan) analyses were carried out to confirm the surface morphology and porous structure of the TeMC materials. Figure 2 shows FE-SEM (Figure 2a,c,e) and HR-TEM (Figure 2b,d,f) images of the as-prepared TeMC materials with different carbonization temperatures. As shown in the SEM images, all of the prepared TeMC materials have an interconnected porous nature with three-dimensional, bicontinuous structures. These structures are more clearly identifiable in the TEM images. As shown, all of the TeMC materials have uniform pore structures that are homogeneously dispersed in the carbon framework.

To understand the porous structure more clearly, nitrogen adsorption–desorption isotherm measurements were carried out. Figure 3a shows the nitrogen sorption isotherms data for the as-prepared TeMC materials; the corresponding porosity properties are summarized in Table 1. The shape of the isotherms can be distinguished into six types according to the International Union of Pure and Applied Chemistry (IUPAC) classification [34]. Among them, the isotherm data of the prepared TeMC materials, as shown in Figure 3a, correspond to type IV with an H1 hysteresis loop, which represents a well-developed mesopore and interconnected pore structure. The specific surface area and pore volumes are calculated and summarized in Table 1. The specific surface areas of the TeMC materials with different carbonization temperatures are 399.68, 455.05, and 526.79 m^2^ g^−1^, and the total pore volumes are 0.80, 0.78, and 0.87 cm^3^ g^−1^. As a result, the specific surface area increased with increasing temperature, and the total pore volume increased accordingly. In particular, the TeMC-900 material has a higher specific surface area and micropore volume than that of other materials (TeMC-700 and TeMC-800), because the carbon surface rapidly decomposed at high temperatures, forming micropores. Therefore, the specific surface area increased with increasing carbonization temperature. Well-developed mesoporous structures and high specific surface areas within the carbon network give rise to more efficient diffusion of electrolyte ions and contact with these ions. Figure 3b shows the wide-angle x-ray diffraction (XRD) spectra of the TeMC materials. All of the materials have two typical broad peaks at 25° and 43°, which were assigned to the [002] and [100] diffraction planes [35]. These XRD peaks indicate that the TeMC materials have an amorphous and disordered carbon structure with many defect sites.

To further investigate the structural characterization of the prepared TeMC samples, Raman spectroscopy was used. As shown in Figure 4, three different peaks were observed at 1350, 1590, and 1510 cm^−1^, corresponding to the *D* (disordered graphitic structure or defective graphite), *G* (graphitic structure), and *A* (amorphous structure) bands. The relative intensity ratio of the *D* to *G* band (*I_D_/I_G_*) is a useful tool for evaluating the number of defects in the carbon framework. As calculated in Figure 4, the *I_D_*/*I_G_* ratio decreased with increasing carbonization temperature due to the decomposition of the defect sites, which are mainly distributed at the edges of the carbon frameworks. As shown in Appendix A, the electrical conductivity of TeMC materials increased with increasing carbonization temperature because of the decrease in the number of defect sites.

To confirm the elemental composition and bonding configurations of the elements in the carbon materials, x-ray photoelectron spectroscopy (XPS) was carried out. As shown in Figure 5 and Table 2, all TeMC materials mainly consisted of carbon, nitrogen, oxygen, and tellurium. After the carbonization process, tellurium was introduced into the carbon structure; XPS analysis confirmed that tellurium was present. As shown in Table 2, according to the increase in carbonization temperature, the contents of Te and N elements were reduced. More specifically, to understand the bonding configuration of the as-prepared carbon materials, a high-resolution XPS spectrum was analyzed. As shown in Appendix A, the C1s spectra (Appendix A) can be deconvoluted into five different peaks at around 283.3, 284.6, 285.7, 286.8, and 288.5 eV, corresponding to C–Te, graphitic carbon (sp^2^ and sp^3^), C–O, C=Te, and C=O, respectively [32]. Deconvolution of the O1s spectra (Appendix A) shows three major peaks at 530.8 (Te–O), 531.8 (C=O), and 533.0 eV (C–O) [36]. The high-resolution N1s XPS spectra (Appendix A) shows four types of nitrogen species at around 398.2 (pyridinic-*N*), 400.1 (pyrrolic-*N*), 401.2 (quaternary-*N*), and 402.4 eV (pyridinic-*N*-oxide) [37]. The Te3d spectra were also deconvoluted into four peaks with binding energies of 573.8, 575.7, 584.2, and 586.2 eV, assigned to the oxidized tellurium (575.7 and 586.2 eV) and the C–Te peak (573.8 and 584.2 eV) [32]. From these high-resolution XPS results, we know that the elemental tellurium is well incorporated into the carbon structure. As shown in Appendix A, the pyridinic and pyrrolic nitrogen are mainly located at the edge side, but quaternary nitrogen is mostly located at the center side. Therefore, when the carbonization temperature is increased, the pyridinic and pyrrolic nitrogen are rapidly decomposed because they are located at the edge side. Appendix A shows elemental mapping data confirming the dispersity of each element in the carbon framework. All elements were homogeneously dispersed in the carbon framework.

Before evaluating the electrochemical performance, TeMC electrodes were prepared by a simple electrospray method. The preparation method was as follows (see Figure 6). First, the as-prepared TeMC materials were homogeneously dispersed in an isopropyl alcohol solution by ultrasonication. Then, this dispersed solution was centrifuged to remove large particles of TeMC materials. The resultant solution was loaded into a syringe with a stainless-steel needle. A high voltage was applied between the needle and substrate. The TeMC materials were directly deposited on the fluorine-doped SnO_2_ (FTO)/glass surface. The loaded amount of the TeMC material was controlled by the electrospray time of the dispersed solution. The preparation method of the electrode is described in more detail in the Appendix A. Generally, the amount of carbon material in the CE greatly affects the catalytic performance, because the active site that can contact the electrolyte has changed. Therefore, to confirm the catalytic performance as a function of the loaded amount of the TeMC material, TeMC electrodes with various loaded amounts (0.1, 0.2, 0.3, and 0.5 mL) were prepared by controlling the electrospray time. The prepared electrodes were denoted as TeMC-*X* (where TeMC is the tellurium-doped, mesoporous carbon, and *X* is the loaded amount of TeMC materials).

To confirm the surface morphology and coverage of the prepared electrodes, SEM measurements were carried out. Figure 7 shows SEM images of the pristine FTO glass, the Pt electrode, and the TeMC electrodes (with the various loaded amounts of TeMC materials). As shown in the SEM image, the platinum nanoparticles of the Pt electrode are uniformly distributed on the FTO glass surface, which can be seen in Figure 7a,b. Figure 7c–f shows the SEM images of the TeMC electrode surfaces. When the amount of sprayed solution increased from 0.1 to 0.5 mL, the amount of TeMC sprayed on the FTO glass also increased, resulting in increased coverage (See Appendix A).

The optical transmittance of the CE is an important factor for bifacial DSSC devices. In addition, the light absorption of the electrolyte is important because illuminated light from the rear side passes through the CE and the electrolyte before finally reaching the photoanode. Therefore, not only the transmittance of the CE, but also the light absorption of the electrolyte, should be considered for enhancing the performance of bifacial DSSC devices. To confirm the transparency of Pt and TeMC CEs, visible transmittance spectra were obtained. Figure 8a shows the transmittance spectra of pristine FTO glass, the Pt CE, and TeMC CEs with different loading amounts (TeMC material was carbonized at 900 °C). The FTO glass has a transmittance of about 80% in the visible light region (400–800 nm). The Pt CE, as a reference, was prepared by a thermal decomposition method; the transmittance was around 75%. In the case of the as-prepared TeMC CEs, the transmittance was lower than that of the Pt CE; as the loaded amount increased, the transmittance decreased significantly. TeMC-0.1 CE, which showed the highest transmittance, exhibited a high transmittance, i.e., about 70%, in the visible light region. As mentioned above, the performance of bifacial DSSCs largely depends on the transmittance of the CE, as well as the light absorption in the electrolyte. Figure 8b shows the absorbance spectra of the cobalt electrolyte. The cobalt electrolyte significantly absorbs light below the wavelength of 400 nm.

To evaluate the electrocatalytic ability of CEs for the cobalt redox mediator [Co(bpy)_3_^2+/3+^], Pt and TeMC CEs were prepared by thermal decomposition and electrospray methods. The symmetrical dummy cells (see Appendix A), with two identical electrodes, were fabricated in order to evaluate the electrode performance. Electrochemical impedance spectroscopy (EIS) is widely used in various fields; it is one of the most useful methods for evaluating electrode performance. First, to optimize the TeMC CEs (different carbonization temperatures), the loaded amount of each TeMC material was fixed (0.5 mL sprayed solution), and symmetrical dummy cells were fabricated. Appendix A shows the EIS data for TeMC CEs with different carbonization temperatures. There are two different semicircles with charge transfer resistances (*R_ct_*) in the high-frequency region and Nernst diffusion impedances (Z_N_) in the low-frequency region. The *R_ct_* is related to the charge transfer resistance at the CE/electrolyte interface, and Z_N_ is the diffusion resistance of the electrolyte between the electrodes (corresponding equivalent circuit, Appendix A). Appendix A shows the EIS results of all TeMC CEs (carbonization temperatures of 700, 800, and 900 °C), exhibiting a lower *R_ct_* value than the Pt CE. Among them, the TeMC CE manufactured at 900 °C had the smallest *R_ct_* value. According to the above results (XPS, Raman, and electrical conductivity), the TeMC-900 material has a low Te doping level and few defect sites, but it has high electrical conductivity. Generally, the CE requires good electrocatalytic properties as well as high electrical conductivity to facilitate the movement of electrons from the external circuit to the electrode surface. For these reasons, the TeMC-900 CE showed better electrocatalytic properties than other TeMC CEs. Thereafter, to make the transparent electrode, the amount of loaded TeMC-900 on the FTO glass was optimized by controlling the electrospray time. When the loaded amount was decreased from 0.5 to 0.1 mL of the dispersed solution of the TeMC-900 material, the *R_ct_* value increased. In the case of 0.08 mL, it had a larger *R_ct_* value than the Pt CE. Thereafter, the electrodes were prepared under these optimized conditions (carbonization temperature: 900 °C, amount of sprayed TeMC solution: 0.1 mL), and were denoted as the TeMC. The role of the CE is to regenerate the oxidized redox species in the electrolyte. The rate of charge transfer from the CE to the electrolyte affects the cathode performance in the DSSC, and can also affect photocurrent generation in the photoanode through dye regeneration by the redox mediator. Therefore, the regeneration rate of the redox mediator by the CE should be comparable to the rate of dye regeneration. The electrocatalytic performance of the CE is determined by the charge transfer resistance (*R_ct_*), which is related to the exchange current density (*J*_0_), as in the following equation (1) [38]:
*J*_0_ = *RT*/*nFR_ct_*(1)
where *R*, *T*, *n*, *F*, and *R_ct_* are the gas constant, temperature, number of electrons involved in the electrochemical reaction of the redox mediator, Faraday’s constant, and charge transfer resistance, respectively. The higher the *J_0_* value, the better the electrocatalytic ability of the CE. Figure 9 shows the EIS and Tafel-polarization plots of the Pt and TeMC dummy cells, and the corresponding *J*_0_ values are summarized in Table 3. The *R_ct_* value of TeMC is 0.49 Ω cm^2^, which is lower than that of Pt (0.62 Ω cm^2^), with corresponding *J*_0_ values of 52.43 and 41.44 mA cm^−2^, respectively. The TeMC CE prepared under optimized conditions showed better electrocatalytic properties than the Pt CE, and it exhibited a transmittance of over 70% in the visible light region (See Figure 8a). It is possible to fabricate bifacial DSSC devices because of these features. In addition, the good electrocatalytic ability of the TeMC CE can be confirmed from the Tafel-polarization data. The value of *J*_0_ is lower than that of the *J*_0_ value calculated from EIS data, but the tendency is well-matched.

To further investigate the electrocatalytic activity of the TeMC CE for the cobalt electrolyte, cyclic voltammetry (CV) was used, and the Pt CE was also studied for performance comparison. As shown in Figure 10, two pairs of redox couples, corresponding to the anodic peak current (*I_pa_*) and cathodic peak current (*I_pc_*), were observed in both Pt and TeMC CEs. These *I_pa_* and *I_pc_* peaks are related to the oxidation of Co(bpy)_3_^2+^ and the reduction of Co(bpy)_3_^3+^ ions, respectively. When evaluating the electrocatalytic ability by CV measurements, the peak current and peak-to-peak distance (*E_pp_*) are important parameters. If the CE has a higher peak current and narrow *E_pp_* values, it has excellent electrocatalytic ability. Even though the Pt and TeMC CEs have similar peak currents, the TeMC CE exhibits good electrocatalytic ability because it has a narrower *E_pp_* value than the Pt CE. The same trend was observed for the CV curves measured at different scan rates (See Appendix A). This CV result was well-matched with the results of the other electrochemical analyses mentioned above.

Based on the electrochemical analysis and various in-depth evaluations of TeMC CEs, the electrocatalytic ability of TeMC CEs was evaluated in an actual DSSC device employing the SM315 [39] sensitizer and the cobalt electrolyte [Co(bpy)_3_^2+/3+^]. Before fabricating the bifacial devices, we first confirmed the current–voltage (J–V) performance of the DSSC with the different TeMC CEs loading amounts. The fabrication method of the DSSC device is precisely described in the Appendix A. Appendix A shows the J–V curves of SM315/ Co(bpy)_3_^2+/3+^ based DSSC devices with Pt and TeMC CEs. The corresponding incident photon-to-current conversion efficiency (IPCE) data of DSSC devices and their photovoltaic performances are summarized in Appendix A. In the dummy cell test, all TeMC CEs (except for TeMC-0.08 CE) had smaller *R_ct_* values than the Pt CE, and the *R_ct_* value increased by reducing the loaded amount of the TeMC material (See Appendix A; In the case of full cell devices, the TeMC-based device has a smaller *R_ct_* value than that of Pt-based device.). Although the *R_ct_* value increased, it could already meet the current generated from the photoanode. Therefore, all devices have similar PCE values (Pt: 12.01%, TeMC-0.1: 12.07%, TeMC-0.2: 12.10%, TeMC-0.3: 12.13%, TeMC-0.5: 12.17%) when fabricating the actual DSSC device (with scattering layer).

To confirm the performance of the bifacial DSSC device, Pt and TeMC CEs were applied to the DSSC device without a scattering layer. Figure 11 shows the J–V curves of bifacial DSSC devices employing the Pt and TeMC CEs and the IPCE data. The detailed photovoltaic performances are displayed in Table 4. The bifacial DSSC devices (see Figure 11a inset) can absorb both incoming light from the front (front irradiation) and rear (rear irradiation) side. Therefore, to use most of the light coming from the rear side, the CE must be transparent, and the electrolyte must absorb the small amount of light. In the case of DSSC performance for front irradiation, the TeMC CE-based device (9.43%) showed a higher power conversion efficiency (PCE) than the Pt CE (9.23%). As shown in Table 4, the short circuit current (*J_SC_*; Pt: 14.58 mA cm^−2^, TeMC: 14.48 mA cm^−2^) and open circuit voltage (*V_OC_*; Pt: 820.51 mV, TeMC: 822.14 mV) values of both devices are similar, but the fill factor (*FF*; Pt: 77.13%, TeMC: 79.23%) values are somewhat different due to the difference in electrocatalytic ability. When irradiated from the rear side, both devices employing Pt and TeMC CEs showed lower PCE values than front irradiated efficiency. The reason for the decrease in the PCE is the low *J_SC_* value, which occurred because light below a wavelength of 500 nm is almost absorbed in the cobalt electrolyte when it is irradiated from the rear side (see Figure 8b). Because of this, the IPCE results (Figure 11b) also showed low conversion efficiency at wavelengths below 500 nm; the light is absorbed in not only the electrolyte, but also the CE. As shown in Figure 8a, Pt and TeMC CEs exhibited transmittances of about 75% and 70%, respectively. In the case of full cells, Pt and TeMC CE-based DSSCs (see Appendix A) gave average transmittances of about 30.76% and 26.53%, respectively. Therefore, the TeMC (8.06%) electrode, with relatively low transmittance, showed a slightly lower PCE performance than the Pt (8.42%) electrode. This is a result of the difference in the amount of light reaching the photoanode, depending on the transmittance of the CEs. Although the TeMC CE had slightly lower rear side irradiation performance than the Pt CE, it showed good performance in both side irradiations due to its high transmittance and electrocatalytic properties.

The electrochemical stability of the CE is one of the most important factors for the practical use of DSSC devices. To confirm the electrochemical stability of the cobalt electrolyte, the dummy cells with Pt and TeMC CEs were freshly prepared. Then, the *R_ct_* value was confirmed by EIS analysis, measured at 0 V from 10^6^ to 10^−1^ Hz. After measuring the *R_ct_* value, the dummy cells were tested through 50 consecutive potential CV cycles from –1 to 1 V at a scan rate of 50 mV s^−1^. These EIS and CV tests were repeated 10 times to confirm the electrochemical stability of the CEs. Figure 12 shows the electrochemical stability test of the symmetrical dummy cells with Pt and TeMC CEs. As shown in Figure 12, the *R_ct_* value of the Pt CE increased significantly from the first to the final cycle. On the other hand, the *R_ct_* value of the TeMC CE increased slightly. Thus, it can be concluded that the TeMC CE has better electrochemical stability than the Pt CE in cobalt electrolytes.

## 4. Conclusions

In summary, a transparent CE for bifacial DSSC devices was prepared using TeMC material. The TeMC had a high doping level (tellurium: 0.21 at.% and nitrogen: 5.74 at.%, from XPS study) and consisted mainly of a mesoporous structure with high electrical conductivity. The TeMC CE itself exhibited high transparency, i.e., about 70% (FTO glass: about 80%, Pt CE: about 75%), while maintaining high degree of electrocatalytic activity and electrical conductivity, as well as improved electrochemical stability. When applied to the bifacial DSSC device, the efficiency of the rear side (TeMC: 8.06%, Pt: 8.42%) was slightly lower than that of the Pt CE, because the transmittance was slightly lower than that of Pt, but the front side (TeMC: 9.43%, Pt: 9.23%) showed better performance due to the good electrocatalytic properties. Based on our results, newly-developed transparent, carbon-based electrodes led to a more stable and effective bifacial DSSC without sacrificing the photovoltaic performance of the device. This technology can be applied to various fields, such as batteries, fuel cells, sensors, etc.

## Figures and Tables

**Figure 1 nanomaterials-10-00029-f001:**
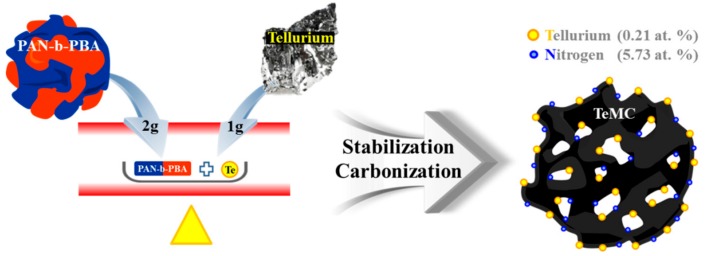
Schematic illustration of the preparation method of tellurium-doped, mesoporous carbon (TeMC) materials.

**Figure 2 nanomaterials-10-00029-f002:**
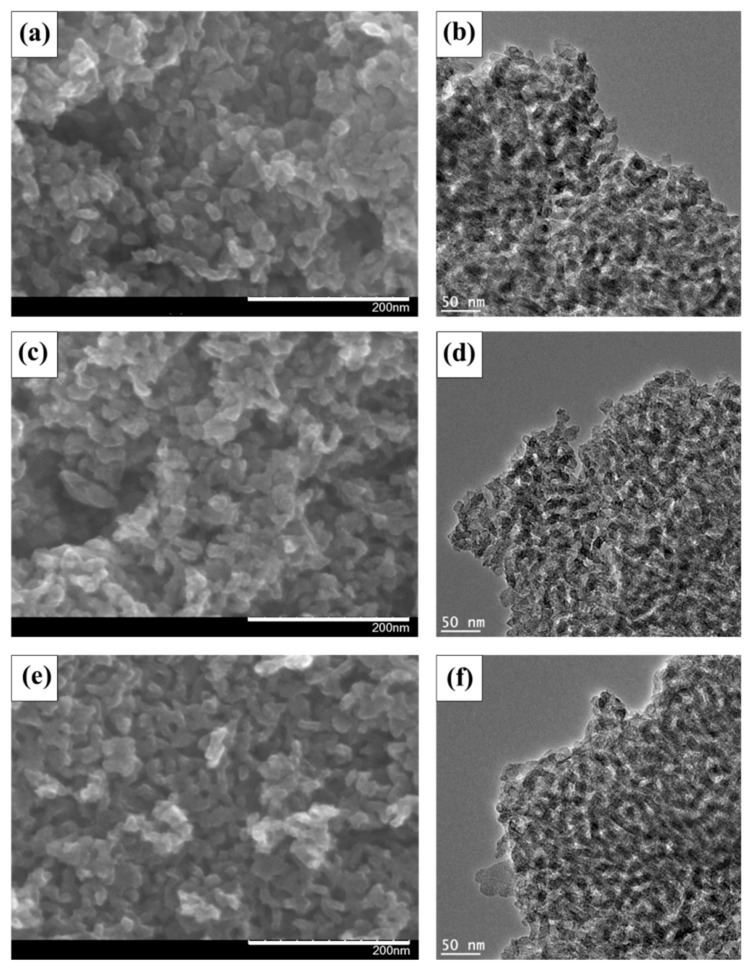
Field emission scanning electron microscopy (**a**,**c**,**e**) and high-resolution transmission electron microscopy (**b**,**d**,**f**) images of the prepared TeMC materials with different carbonization temperatures (**a**,**b**): 700 °C; (**c**,**d**): 800 °C; (**e**,**f**): 900 °C.

**Figure 3 nanomaterials-10-00029-f003:**
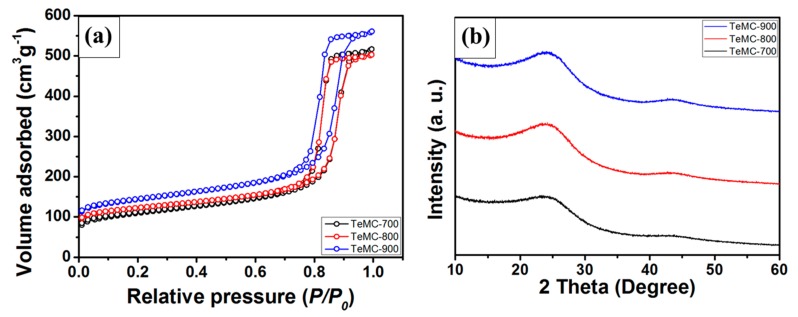
(**a**) Nitrogen adsorption–desorption isotherm data and (**b**) wide-angle x-ray diffraction spectra of TeMC materials.

**Figure 4 nanomaterials-10-00029-f004:**
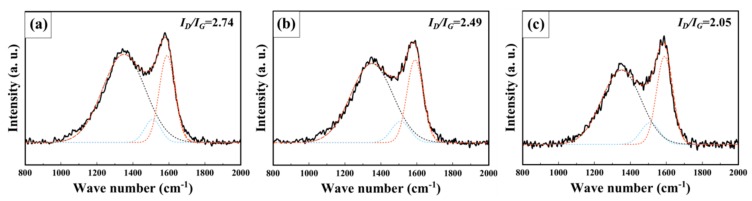
Raman spectra of TeMC materials with different carbonization temperatures (**a**): 700, (**b**): 800, (**c**): 900 °C.

**Figure 5 nanomaterials-10-00029-f005:**
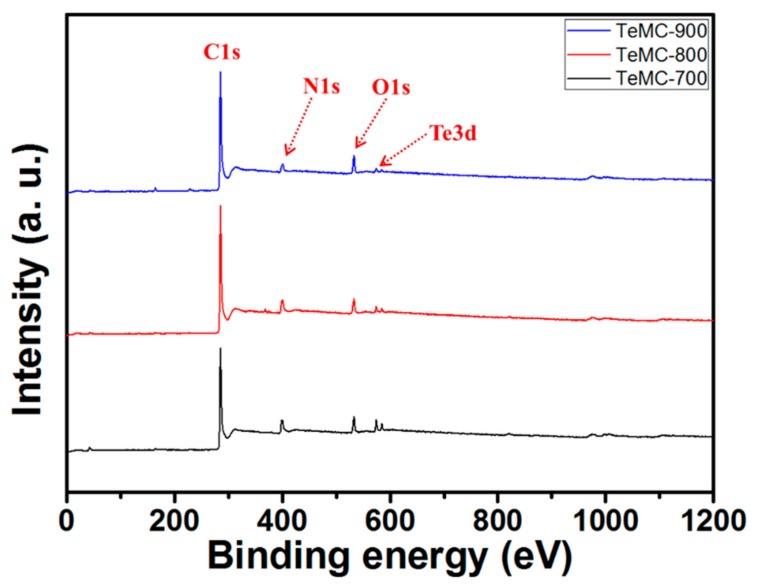
X-ray photoelectron spectroscopy spectra of TeMC materials with different carbonization temperatures.

**Figure 6 nanomaterials-10-00029-f006:**
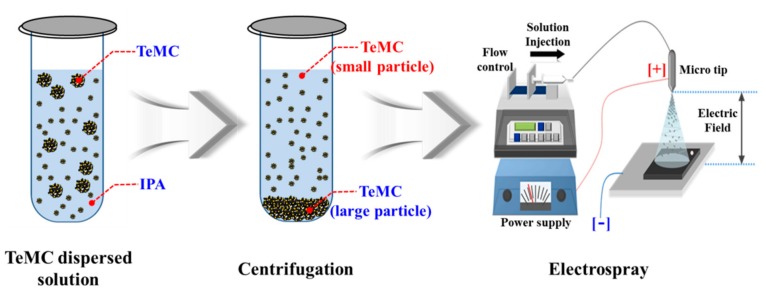
Preparation method of TeMC electrodes via the electrospray technique.

**Figure 7 nanomaterials-10-00029-f007:**
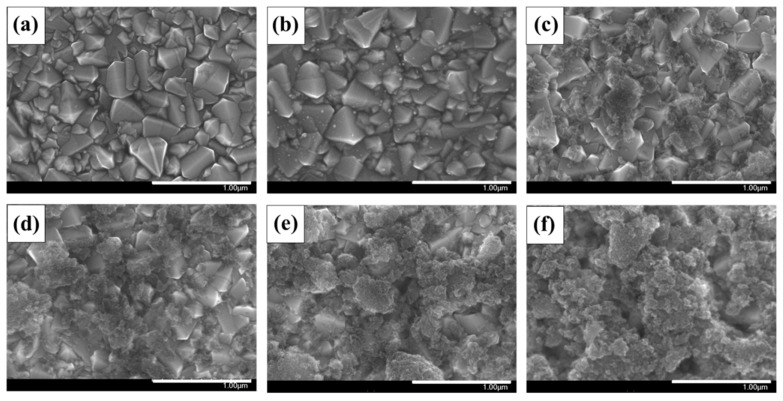
Scanning electron microscopy images of the pristine FTO glass (**a**), Pt electrode (**b**), TeMC electrodes (**c**): TeMC-0.1, (**d**): TeMC-0.2, (**e**): TeMC-0.3, (**f**): TeMC-0.5.

**Figure 8 nanomaterials-10-00029-f008:**
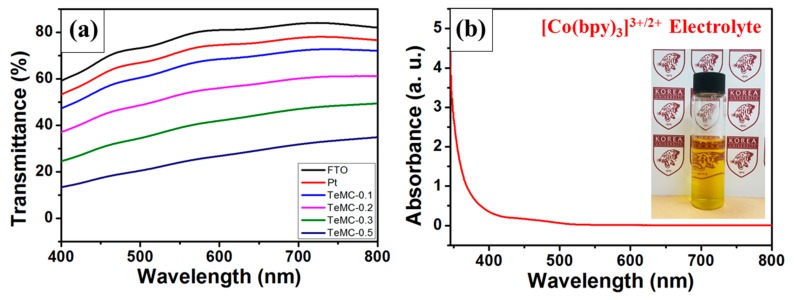
(**a**) Transmittance spectra of pristine FTO glass, the Pt CE, and the TeMC CEs (different loading amounts of TeMC materials). (**b**) Absorption spectra of the cobalt electrolyte, which was diluted 100 times.

**Figure 9 nanomaterials-10-00029-f009:**
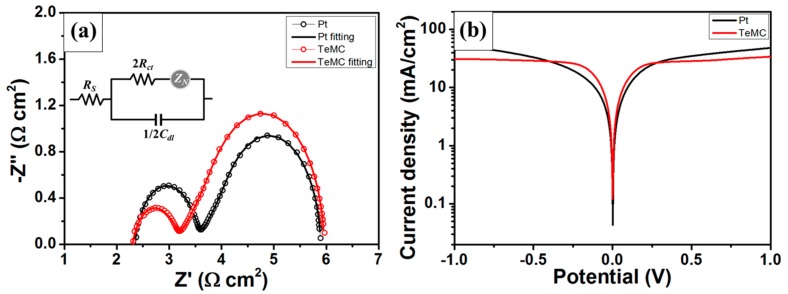
(**a**) Nyquist plots and (**b**) Tafel-polarization plots of the Pt and TeMC dummy cells with the Co(bpy)_3_^2+/3+^ electrolyte.

**Figure 10 nanomaterials-10-00029-f010:**
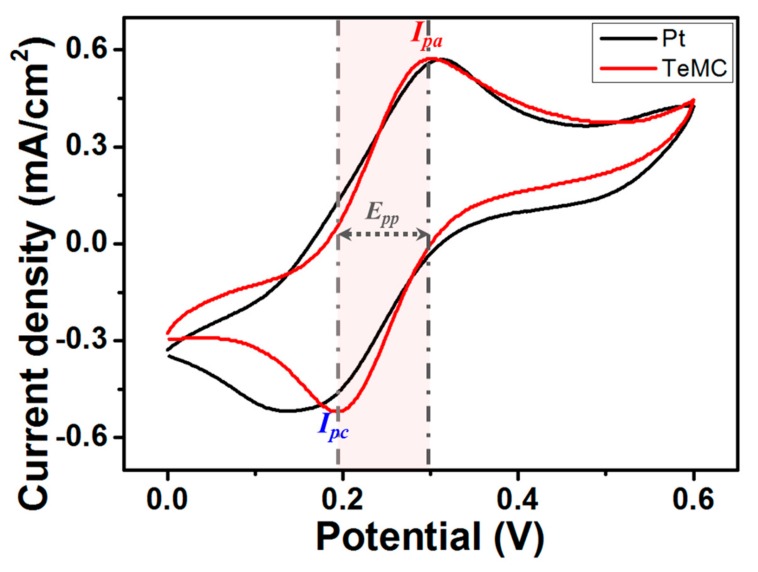
Cyclic voltammetry curves for the Co(bpy)_3_^2+/3+^ redox couples obtained at a scan rate of 50 mV s^−1^ using the Pt and TeMC CEs as the working electrode, a Pt wire as the CE, Ag/AgCl as the reference electrode, and 0.1 M LiClO_4_ as the supporting electrolyte.

**Figure 11 nanomaterials-10-00029-f011:**
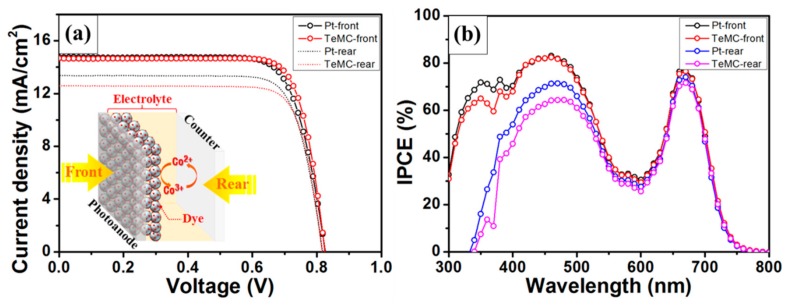
(**a**) *J–V* curves of bifacial DSSC devices with Pt and TeMC CEs under one sun illumination (AM 1.5 G). The inset is a schematic illustration of the bifacial DSSC device. (**b**) The corresponding IPCE data of the bifacial DSSC devices.

**Figure 12 nanomaterials-10-00029-f012:**
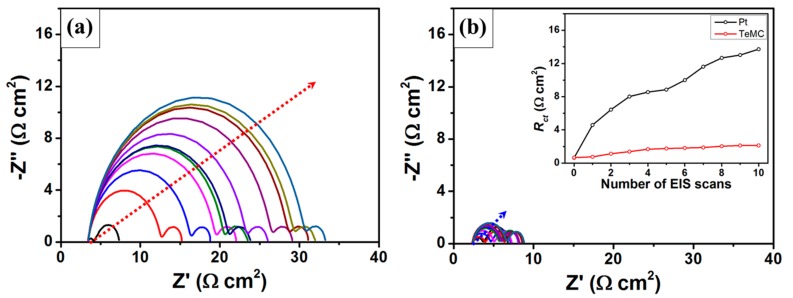
Electrochemical stability test of symmetrical dummy cells with Pt (**a**) and TeMC CEs (**b**). The sequence of measurements is as follows: EIS was measured at 0 V from 10^6^ to 10^−1^ Hz. Then, CV scans of 50 cycles were performed from −1 to 1 V at a scan rate of 50 mV s^−1^. The sequence of the electrochemical test was repeated 10 times. The inset is the *R_ct_* value change of dummy cells with Pt and TeMC CEs.

**Table 1 nanomaterials-10-00029-t001:** Porosity properties of TeMC materials with different carbonization temperatures.

Sample	*S*_BET_ (m^2^ g^−1^)	*V*_total_ (cm^3^ g^−1^)	*V*_micro_ (cm^3^ g^−1^)	*V*_meso_ (cm^3^ g^−1^)	Pore Size (nm)
TeMC-700	399.68	0.80	0.08	0.73	12
TeMC-800	455.05	0.78	0.11	0.68	12
TeMC-900	526.79	0.87	0.12	0.75	11

*S*_BET_: BET surface area, *V*_total_: total pore volume, *V*_micro_: micropore volume, *V*_meso_: mesopore volume.

**Table 2 nanomaterials-10-00029-t002:** Elemental composition as determined by x-ray photoelectron spectroscopy.

Sample	Element (at.%)			
	C	O	N	Te
TeMC-700	83.63	5.80	10.01	0.55
TeMC-800	86.34	5.04	8.36	0.26
TeMC-900	87.47	6.40	5.73	0.21

**Table 3 nanomaterials-10-00029-t003:** Electrochemical parameters of the CE in the symmetrical dummy cells by LSV (linear sweep voltammetry) and EIS (electrochemical impedance specteoscopy).

		LSV		EIS			
CE	Electrolyte	*R_CV_* (Ω cm^2^)	*J_o_* (mA cm^−2^)	*R_s_* (Ω cm^2^)	*R_ct_* (Ω cm^2^)	*C_dl_* (μF cm^2^)	*J_o_* (mA cm^−2^)
Pt	Co(bpy)_3_^2+/3+^	9.37	2.74	2.30	0.62	14.38	41.44
TeMC		6.87	3.74	2.17	0.49	28.91	52.43

**Table 4 nanomaterials-10-00029-t004:** Photovoltaic performance of DSSCs.

Sample	Dye/Electrolyte	*J_SC_* (mA cm^−2^)	*V_OC_* (mV)	*FF* (%)	PCE (%) *
Pt-front	SM315/Co(bpy)_3_^2+3+^	14.58 ± 0.11	820.51 ± 4.86	77.13 ± 0.57	9.23 ± 0.07
TeMC-front		14.48 ± 0.12	822.14 ± 3.73	79.23 ± 0.23	9.43 ± 0.10
Pt-rear		13.25 ± 0.14	817.00 ± 1.43	77.78 ± 0.10	8.42 ± 0.08
TeMC-rear		12.52 ± 0.11	813.38 ± 3.04	79.13 ± 0.25	8.06 ± 0.09

* The average power conversion efficiency (PCE) was calculated in five different cells, measured under AM 1.5 G illumination (100 mW cm^−2^).

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
