# Peer review of "Tellurium-Doped, Mesoporous Carbon Nanomaterials as Transparent Metal-Free Counter Electrodes for High-Performance Bifacial Dye-Sensitized Solar Cells"

_nanomaterials, 2019, doi:10.3390/nano10010029_

Round 1

Reviewer 1 Report

The authors reported the preparation of tellurium-doped mesoporous carbon (TeMC) nanomaterials, which were used to fabricate a transparent counter electrode (CE) for bifacial dye-sensitized solar cells (DSSCs). The reported photovoltaic efficiency under front and rear side irradiation are 9.43% and 8.06%, respectively. This work is interesting and would be helpful to DSSC and some other applications.

Below are some issues for the authors:

Figure 8(a) shows the transmittance spectra for TeMC CEs with different loading amounts. Please specify under what carbonization temperature were these TeMC materials prepared. Please provide the thicknesses for TeMC layers deposited on the FTO glass with different loading amounts of TeMC materials. In lines 244-246, the author wrote, "When the loaded amount was decreased from 0.5 to 0.1 mL of the dispersed solution of the TeMC-900 material, the Rct value increased." Why? In this work, it is found that the TeMC CE has a better electrochemical stability than that of the Pt CE. Can the author provide the reason? The authors claim that the photovoltaic efficiency of the bifacial DSSC with the TeMC CE is the highest. It is desirable to provide comparison of this work to other recent works, so that the readers can understand the advance of the bifacial DSSCs.

Author Response

Reviewer 1:

The authors reported the preparation of tellurium-doped mesoporous carbon (TeMC) nanomaterials, which were used to fabricate a transparent counter electrode (CE) for bifacial dye-sensitized solar cells (DSSCs). The reported photovoltaic efficiency under front and rear side irradiation are 9.43% and 8.06%, respectively. This work is interesting and would be helpful to DSSC and some other applications.

Below are some issues for the authors:

Figure 8(a) shows the transmittance spectra for TeMC CEs with different loading amounts. Please specify under what carbonization temperature were these TeMC materials prepared.

Answer: In line 215, we have given the specified carbonization temperature in the revised manuscript.

Please provide the thicknesses for TeMC layers deposited on the FTO glass with different loading amounts of TeMC materials.

Answer: Actually, when we prepared the TeMC CE, that TeMC material was dispersed on the FTO glass. It is not like a layer, but just deposited the FTO glass. Thus, it is difficult to measure the thickness.

In lines 244-246, the author wrote, "When the loaded amount was decreased from 0.5 to 0.1 mL of the dispersed solution of the TeMC-900 material, the Rct value increased." Why?

Answer: When the loaded amount was decreased, the active sites of CE were decreased according to the decrease of loaded amount. Therefore, Rct value was increased.

In this work, it is found that the TeMC CE has a better electrochemical stability than that of the Pt CE. Can the author provide the reason?

Answer: The nanoparticle Pt-based CEs generally suffer from surface oxidations and aggregation in electrolytes. However, there are almost no aggregation and oxidations for carbon materials. Thus, the TeMC CE has a better electrochemical stability than that of the Pt CE.

The authors claim that the photovoltaic efficiency of the bifacial DSSC with the TeMC CE is the highest. It is desirable to provide comparison of this work to other recent works, so that the readers can understand the advance of the bifacial DSSCs.

Answer: We added the comparison chart of reported performances for bifacial DSSCs in Table S2.

Reviewer 2 Report

This work from Kim and coworkers describes the fabrication of transparent counter electrodes for DSSCs. The aim is to improve electrocatalytic activity and electrical conductivity, using tellurium-doped mesoporous carbon (TeMC) materials . Counter electrode materials are highly imporant for the improvement of DSSC performances and attract significant attention within the materials community. Thus the topic is very suitable for this journal. The paper is well written and appears sound in terms of content.

A very important point that needs to be addressed is that it does not appear that the DSSCs are masked for the measurements. This means that all measurements are overestimates (see Snaith Energy Environ. Sci., 2012,5, 6513). Ideally, the data should be repeated with masked DSSCs. At the very least, the authros must state that they have worked with unmasked devices and that the Jsc values (and therefore PCEs) are probably overestimated.  

Revisions suggested:

Use of tellurium on a commercial scale cold be problematic. Although the element itself has a low toxicity, it has unpleasant side effects to the human system. Have the authors considered this when they chose this dopant? A comment should be made. For example on line 82, there is no precautionary note when 1g of Te powder is 'physically mixed ' with PAN-b-PBA copolymer.

Line 38 Use of 'electrolyte' should incorporate all components, or the authors need to replace by 'redox couple' since they are referring only to the redox couple (shuttle) and not the whole electrolyte. 

Line 97 N2  subscript 2

Figure 1 right: Could imply a defined Te:N ratio. Please clarify in the caption.

Figure 2. The 200 nm scale bars are not clear.  Also in Figure 7. A single bar would be unambiguous.

Line 126  m2 g-1  , Line 163 sp2 and sp3    superscripts

On line 207 the authors correctly point to problems of light absorption of the electrolyte. This is a well known problem with triiodide because it is rather strongly coloured, and the cobalt-based electrolyte also absorbs rather strongly. It is not clear to me if this is a criterion for th choice of the Co2+/3+ couple or whether the choice has been made in terms of redox potential. Please clarify in the text.

EIS; The Nyquist plots in Figure S7 and Figure 9 do not show fitting. Please show both the expermental plots and the fits.

Supp. Data. Experimental. The formula Co(bpy)3(BCN4)2 is incorrect. Change to [Co(bpy)3][B(CN)4]2

Author Response

Reviewer 2:

This work from Kim and coworkers describes the fabrication of transparent counter electrodes for DSSCs. The aim is to improve electrocatalytic activity and electrical conductivity, using tellurium-doped mesoporous carbon (TeMC) materials. Counter electrode materials are highly imporant for the improvement of DSSC performances and attract significant attention within the materials community. Thus the topic is very suitable for this journal. The paper is well written and appears sound in terms of content.

A very important point that needs to be addressed is that it does not appear that the DSSCs are masked for the measurements. This means that all measurements are overestimates (see Snaith Energy Environ. Sci., 2012,5, 6513). Ideally, the data should be repeated with masked DSSCs. At the very least, the authros must state that they have worked with unmasked devices and that the Jsc values (and therefore PCEs) are probably overestimated.

Answer: All the DSSCs have been masked for the measurements. The mask with an aperture of 0.141cm2 is used to define the active area of the test cells.

Revisions suggested:

Use of tellurium on a commercial scale could be problematic. Although the element itself has a low toxicity, it has unpleasant side effects to the human system. Have the authors considered this when they chose this dopant? A comment should be made. For example on line 82, there is no precautionary note when 1g of Te powder is 'physically mixed ' with PAN-b-PBA copolymer.

Answer: We mixed the copolymer and the Te powder in a closed system with strict protective furnace, thus, it is less dangerous.

Line 38 Use of 'electrolyte' should incorporate all components, or the authors need to replace by 'redox couple' since they are referring only to the redox couple (shuttle) and not the whole electrolyte.

Answer: Thanks for pointing out this error. We have corrected it as suggested.

Line 97 N2 subscript 2

Answer: We have corrected it.

Figure 1 right: Could imply a defined Te:N ratio. Please clarify in the caption.

Answer: We have added the Te : N ratio in Figure 1. It is obtained from XPS data.

Figure 2. The 200 nm scale bars are not clear. Also in Figure 7. A single bar would be unambiguous.

Answer: We have corrected it.

Line 126 m2 g-1, Line 163 sp2 and sp3 superscripts

Answer: We have corrected it.

On line 207 the authors correctly point to problems of light absorption of the electrolyte. This is a well known problem with triiodide because it is rather strongly coloured, and the cobalt-based electrolyte also absorbs rather strongly. It is not clear to me if this is a criterion for the choice of the Co2+/3+ couple or whether the choice has been made in terms of redox potential. Please clarify in the text.

Answer: This figure shows absorbance spectra of iodine and cobalt redox electrolytes. As seen in this figure, iodine electrolyte has strong absorption around 370 nm. Therefore, if iodine electrolyte applied for the bifacial DSSC device, rear side DSSC performance will be decreased due to its strong absorption (ACS Appl. Mater. Interfaces 2018, 10, 8611−8620).

EIS; The Nyquist plots in Figure S7 and Figure 9 do not show fitting. Please show both the expermental plots and the fits.

Answer: We added the fitting data of Pt and TeMC CEs in Figure 9a using the inserted equivalent circuit.

Supp. Data. Experimental. The formula Co(bpy)3(BCN4)2 is incorrect. Change to [Co(bpy)3][B(CN)4]2

Answer: We have corrected it as suggested.

Reviewer 3 Report

Authors should subjoin the stability experiments of the counter electrode , this is an important experiment for the electrode.

The magnification of Figure 7-(d) is different from others, please correct it.

In line 258 authors describe “the higher the J0 value, the better the electrocatalytic ability of the CE” please describe and explain more clearly.

Table 4 is important data. It would be more convincing if authors can compare their results with more relevant literature.

In this study used the SM315 dye. Please describe it advantages and disadvantages.

Please explain the IPCE measurement for the wavelength of 300 nm~ 350 nm, why the efficiency of the rear measurement result is too bad.

The Nyquist plots shown in Table 3 are displayed in the supporting Information and will confuse the reader with the measurement in Figure 9. Please correct it.

Please add the EIS measurement equivalent circuit in the manuscript.

Author Response

Reviewer 3:

Authors should subjoin the stability experiments of the counter electrode, this is an important experiment for the electrode.

Answer: We have already confirmed the stability test of counter electrode in dummy cell. Figure 12 shows stability data for Pt- and TeMC- CE.

The magnification of Figure 7-(d) is different from others, please correct it.

Answer:: We have corrected it.

In line 258 authors describe “the higher the J0 value, the better the electrocatalytic ability of the CE” please describe and explain more clearly.

Answer: The performance of cathode in DSSC is determined by the electrocatalytic reduction ability of redox mediator at the CE, which ultimately affects dye regeneration and photocurrent generation. And the rate of reduction of redox mediator should be comparable to dye regeneration rate, which is defined by the photocurrent density (Jsc). Therefore, to prevent the electron loss at the CE, the exchange current density (J0) at the CE should be comparable to the photocurrent density (Jsc).

Table 4 is important data. It would be more convincing if authors can compare their results with more relevant literature.

Answer: We added the comparison chart of reported performances for bifacial DSSCs in Table S2.

In this study used the SM315 dye. Please describe it advantages and disadvantages.

Answer: As far as we can see, SM315 is one of the best sensitizers which can achieve extremely high efficiency of 13.1 % by using cobalt electrolyte. The disadvantage is that SM315 is not commercialized, thus, we need to synthesize it.

Please explain the IPCE measurement for the wavelength of 300 nm~ 350 nm, why the efficiency of the rear measurement result is too bad.

Answer: We have already explained this in line 318.

The reason for the decrease in the PCE is the result of the low JSC value, which is because light below a wavelength of 500 nm is almost absorbed in the cobalt electrolyte when the light is irradiated from the rear side.

The Nyquist plots shown in Table 3 are displayed in the supporting Information and will confuse the reader with the measurement in Figure 9. Please correct it.

Answer: The Nyquist plots shown in supporting information (Fig. S7) showed the EIS data for TeMC CEs with different carbonization temperatures and different loaded amount. But the Figure 9a is the Nyquist plots of Pt and TeMC dummy cells in the optimized condition (carbonization temperature: 900℃, amount of sprayed TeMC solution: 0.1 mL).

Please add the EIS measurement equivalent circuit in the manuscript.

Answer: We added the equivalent circuit in Figure 9.

Round 2

Reviewer 1 Report

None.

Author Response

None.

Reviewer 2 Report

This revised paper is improved. However, in answer to my question about cell masking, the authors say:

Answer: All the DSSCs have been masked for the measurements. The mask with an aperture of 0.141cm2 is used to define the active area of the test cells.

However, this statement is not made in the manuscript and should be. The supporting information describes cell fabrication and measuring methods, but there is no mention that the cells are masked. Please add this statement. 

---

In the Supporting information, (BCN4)2 is incorrect and should be [B(CN)4]2 throughout.

Author Response

Reviewer2:

This revised paper is improved. However, in answer to my question about cell masking, the authors say:

Answer: All the DSSCs have been masked for the measurements. The mask with an aperture of 0.141cm2 is used to define the active area of the test cells.

However, this statement is not made in the manuscript and should be. The supporting information describes cell fabrication and measuring methods, but there is no mention that the cells are masked. Please add this statement. 

Answer: Now we have added this statement to the Supporting information.

In the Supporting information, (BCN4)2 is incorrect and should be [B(CN)4]2 throughout.

Answer: Thanks for pointing out this error. We have corrected it as suggested.